# Investigating the Impact of Dietary Fibers on Mycotoxin Bioaccessibility during In Vitro Biscuit Digestion and Metabolites Identification

**DOI:** 10.3390/foods12173175

**Published:** 2023-08-23

**Authors:** Rosalía López-Ruiz, Jesús Marin-Saez, Sara. C. Cunha, Ana Fernandes, Victor de Freitas, Olga Viegas, Isabel M. P. L. V. O. Ferreira

**Affiliations:** 1LAQV/REQUIMTE, Laboratory of Bromatology and Hydrology, Department of Chemical Sciences, Porto University, 4050-313 Porto, Portugal; jms485@ual.es (J.M.-S.); sara.cunha@ff.up.pt (S.C.C.); olgaviegas@fcna.up.pt (O.V.); isabel.ferreira@ff.up.pt (I.M.P.L.V.O.F.); 2Research Group “Analytical Chemistry of Contaminants”, Department of Chemistry and Physics, Research Centre for Mediterranean Intensive Agrosystems and Agri-Food Biotechnology (CIAIMBITAL), University of Almeria, Agrifood Campus of International Excellence, ceiA3, E-04120 Almeria, Spain; 3LAQV/REQUIMTE, Department of Chemistry and Biochemistry, Science Faculty, Porto University, 4169-007 Porto, Portugal; ana.fernandes@fc.up.pt (A.F.); vfreitas@fc.up.pt (V.d.F.); 4Faculty of Nutrition and Food Sciences, University of Porto, 4150-180 Porto, Portugal

**Keywords:** mycotoxins, dietary fibers, in vitro digestion, mass spectrometry, bioaccessibility, liquid chromatography, metabolites, non-targeted analysis

## Abstract

Mycotoxins contamination is a real concern worldwide due to their high prevalence in foods and high toxicity; therefore, strategies that reduce their gastrointestinal bioaccessibility and absorption are of major relevance. The use of dietary fibers as binders of four mycotoxins (zearalenone (ZEA), deoxynivalenol (DON), HT-2, and T-2 toxins) to reduce their bioaccessibility was investigated by in vitro digestion of biscuits enriched with fibers. K-carrageenan is a promising fiber to reduce the bioaccessibility of ZEA, obtaining values lower than 20%, while with pectin a higher reduction of DON, HT-2, and T-2 (50–88%) was achieved. Three metabolites of mycotoxins were detected, of which the most important was T-2-triol, which was detected at higher levels compared to T-2. This work has demonstrated the advantages of incorporating dietary fibers into a biscuit recipe to reduce the bioaccessibility of mycotoxins and to obtain healthier biscuits than when a conventional recipe is performed due to its high content of fiber.

## 1. Introduction

Mycotoxins are natural toxins produced by several fungus families, of which the most important are *Fusarium*, *Aspergillus*, *Penicillium*, and *Alternaria* [1]. Despite the existence of strict regulations in several countries to prevent mycotoxins contamination, these contaminants can be found in a wide range of food products, and it is expected to rise with climate change as it benefits the growth of fungi [2]. In addition to the contamination of food crops, mycotoxins can also reach the food chain through food of animal origin [3]. Therefore, a relatively low to occasionally high level of these compounds might be found in the diet [4,5].

One of the most important mycotoxin families is trichothecene mycotoxins. It includes more than 180 compounds; although, deoxynivalenol (DON), which belongs to the trichothecene B toxin, and the HT-2 and T-2 toxins, which belong to the trichothecene A toxin, are the compounds with major health concerns [6]. DON is the most widespread mycotoxin in human food, being present in more than 80% of food commodities around the world [7]. Potential hazardous levels of trichothecene can occur naturally in moldy grains, cereals, and agricultural products. Trichothecene toxins are produced in a wide range of temperatures, even in the snow [8]. In addition, they cause severe health effects such as growth retardation, myelotoxicity, hematotoxicity, or necrotic lesions [9,10]. Finally, although zearalenone (ZEA) is not a trichothecene mycotoxin, it is usually found in several cereal grains and, as it is structurally analogous to estrogen, poses estrogenic effects, in addition to hepatotoxic, hematotoxic, immunotoxic, genotoxic, teratogenic, and carcinogenic effects, on different animal species [10,11].

The oral route is considered one of the most significant pathways of human exposure to mycotoxins, the gastrointestinal tract is continuously exposed to these toxins [12]. However, to exert any deleterious effect, mycotoxins must be available, which means that they must be released from the food matrix and become bioaccessible to be absorbed across epithelial cells and finally transported to target tissues [13,14].

Various treatments are available based on chemical, physical, or microorganism technology to decrease the presence of mycotoxins in cereals before or after harvest. For example, chemical decontamination is the use of sodium metabisulfite (Na_2_S_2_O_5_), which can reduce the contamination of cereal grains with DON by forming a sulfonated derivative of DON. Additionally, some agents such as adsorbents, health stimulants, and detoxifiers have previously been used to mitigate mycotoxins, at least in feed samples. These agents mitigate toxic effects by reducing absorption, enhancing immune functions, and acting as detoxifiers. However, for DON there is not a good enough mitigation methodology and some of the used ones were not cost-effective, or they were only proposed for animal use. In addition, information is not available for some mycotoxins such as T-2 toxins [15]. Other approaches to detoxification include physical methods, such as UV treatment, the treatment of cereal grains with ozone or cold atmospheric pressure plasma (CAPP) with ambient air as the working gas [10,16], or the addition of adsorption agents. Adsorption agents may be mycotoxin binders derived from minerals, such as bentonite, montmorillonite, zeolites, or activated charcoal, or from micronized plant-based lignocellulosic materials. Another option is the use of microorganisms that can degrade specific mycotoxins in food, such as lactic acid bacteria that reduce mycotoxins biosynthesis [10]. In addition, in the last year (2022), new strategies for controlling mycotoxins in foods are based on the use of natural products as phenolic compounds because they can inhibit the mycotoxin production through their antioxidant properties, downregulation of the gene’s expression, and introduced structural modifications of the fungal membrane [10]. For all of that, new, cheaper, and tested human food mitigation strategies need to be developed.

In the last few years, several investigations have assessed the bioaccessibility of mycotoxins to gather information on their release from different food matrices during in vitro digestion. Most studies have focused on specific matrices (for example, cereals, meat, and fish) [17,18,19], showing that the food matrix influences the final bioaccessibility of mycotoxins, but did not study their release on a processed food approach, which will certainly have a different impact on their bioaccessibility. In this sense, as the use of binders emerged, various substances have been proposed as mycotoxin binders for animal feed purposes, such as activated carbons, aluminosilicate-based products, and complex indigestible carbohydrates [4,20,21,22]. The latter are dietary fibers that could be used as binders to efficiently counteract the toxic effects of mycotoxins in human foods. The literature has little experimental evidence on the in vitro ability of dietary fibers to act as mycotoxins binders during digestion. To our knowledge, there are only two publications that relate mycotoxins to dietary fibers [14] or fiber-rich food sources [23], but none of these studies evaluated the impact of the incorporation of processed food, which is an approach that is a more realistic situation. There exists a wide range of fibers with distinct physicochemical properties (e.g., linear or with a variable degree of branching or negatively or positively charged) that may impact mycotoxin release differently, mostly considering the gradual acidification of the stomach.

The present study aims to assess the impact of dietary fibers on the bioaccessibility of mycotoxins (DON, ZEA, HT-2, and T-2) in biscuits prior to in vitro digestion and the release of bioaccessibility from the chyme to the duodenum, which probably reduces their bioaccessibility and consequently influences their intestinal absorption. To test that, biscuits were prepared, adding commercial dietary fibers such as arabinogalactan, pectin, and k-carrageenan. Dietary fiber from apple pomace (7–15% pectin) was also tested. For the separation and detection of DON, ZEA, HT-2, and T-2 mycotoxins, liquid chromatography coupled to a triple quadrupole mass spectrometer was used.

## 2. Materials and Methods

### 2.1. Materials and Reagents

The internal standards for mycotoxins DON, HT-2, T-2, ZEA, and OTA-d5 (IS) (> 98% of purity for all) were purchased from Sigma-Aldrich (St. Louis, MO, USA). ^13^C_15_-DON IS solution was obtained from Fluka (West Chester, PA, USA).

For the extraction of mycotoxins, ACN and *n*-hexane were purchased from Sigma-Aldrich. Ultrapure water (0.054 μS/cm) was supplied by a “Seral” system *SeralPur Pro 90* CN (Ritterhude, Germany). Finally, porcine α-amylase, pepsin, bile, and pancreatin enzymes were used for in vitro digestion and provided by Sigma-Aldrich.

#### Dietary Fibers Employed for Biscuits Preparation

Dietary fibers such as arabinogalactan from larch wood, pectin from apple (poly-D-galacturonic acid methyl ester), and k-carrageenan from plants were purchased from Sigma-Aldrich. Apple flour, obtained as a by-product of apple juice production (Bravo de Esmolfe (*Malus domestica Borkh*), was kindly provided by Sumol + Compal^®^ and processed in the laboratory as indicated in previous work [24]. After being thawed at a temperature of 4 °C for a duration of 24 h, the leftover apple material underwent a washing process using cold water (4 L of water for every 1 kg of pomace). The intention was to eliminate any free sugars. This mixture was then filtered using a cloth filter and subsequently manually pressed to eliminate excess water [25,26]. Subsequently, the remaining residue was frozen at a temperature of −20 °C and subjected to lyophilization for a minimum of 48 h, referred to as FP. The dried apple residue was also utilized in its dehydrated state, denoted as FS. Both FS and FP were subjected to grinding in a household grinder in order to obtain a uniform apple flour. Due to insufficient homogeneity in the case of FP, an additional step of powdering was carried out. The resulting flours were hermetically sealed in plastic bags and stored at a temperature of −20 °C for later use. The moisture and fiber (both soluble and total) content were assessed in accordance with the methodology outlined in Martins et al. [27] utilizing commercially available kits (K-TDFR, Megazyme, Cork, Ireland). The sugar content was determined according to Santos et al. [28] (Appendix A).

### 2.2. Biscuit Recipe and Preparation

The ingredients for the biscuits were purchased at the local markets of Porto (Portugal). The recipe was prepared according to the procedure “*salt and water biscuit*” and using the same procedure described by López-Ruiz et al. [24]; the following ingredients were mixed: 500 g of T-45 flour, 2 g of NH_4_HCO_3_, 9 g of NaHCO_3_, 136 g of sugar, 4 g of NaCl, 20 g of malt extract, 100 g of glucose-fructose syrup, and 120 g of sunflower oil. Later, they were all introduced in a mixer and mashed continuously for 10 min. Once the dough was homogeneous, 5% of the dietary fibers were added and mashed again for 5 min to homogenize the mixture. To test the impact of mycotoxins in this step, the mixture was spiked with a known concentration of each mycotoxin (500 µg/kg). The biscuits shape was formed (approximately 25 g of dough per biscuit) by employing a circular mold to ensure that all biscuits have the same dimensions, and they were baked in an oven chamber on the middle shelf (out of seven) for 12 min at 190 °C. Finally, the biscuits were kept for 3 h at room temperature to ensure that they were tempered. Later, they were crushed, homogenized, and introduced into a plastic pot prior to extraction and analysis. It is important to mention that the corresponding amount of dietary fiber added to the biscuit’s recipe was subtracted from the amount of flour added. Therefore, when K, A, or PC was added (25 g), the total amount of flour in the recipe was 475 g. For FS (350 g), the flour was 150 g, and for FP (157 g), the flour was 343 g. In addition, with FS, glucose-fructose syrup and sugar were not added because the pomace provided enough sugar (Table 1).

### 2.3. Sample Treatment

#### 2.3.1. Mycotoxin Extraction

The extraction was based on a solid liquid extraction method (SLE) developed in our research group for human blood serum samples [29] without the use of acid in the extraction solvents and increasing the amount of sample and the extraction solvent volume. However, the extraction solvent ratio (ACN/water) was the same (80:20, *v*/*v*). Briefly, 2 g of biscuit was placed in a 50 mL Falcon tube and extracted with 10 mL of ACN:water (80:20, *v*/*v*). Samples were shaken for 10 min on a rotatory shaker and 10 min in an ultrasonic bath. The samples were centrifuged for 8 min at 7500 r.p.m (8170× *g*) at 0 °C. An amount of 4 mL of supernatant was placed in a 15 mL Falcon tube and 4 mL of n-hexane was added. The tube was shaken for 1 min in a vortex and centrifuged for 8 min at 7500 r.p.m (8170× *g*) at 0 °C. An amount of 1 mL of the supernatant was placed in a vial and concentrated to 0.25 mL under a gentle stream of nitrogen. The samples were reconstituted with 0.23 mL of organic mobile phase (methanol/water, 98:2 (*v*/*v*) and 5 mM ammonium acetate), and 20 µL of a solution of 1 mg/L of ^13^C_15_-DON IS was added prior to LC-MS analysis.

#### 2.3.2. In Vitro Digestion

The in vitro biscuits digestion procedure was based on the internationally standardized method (INFOGEST) described by Brodkorb et al. [30] and López-Ruiz et al. [24]. Three replicates were prepared for each sample, and an additional blank sample was included for the purpose of pH adjustment during digestion and as a baseline matrix for analysis. To create the biscuits sample mixture, 2 g of the sample was combined with 1.6 mL of simulated salivary fluid (SSF), along with 0.2 mL of α-amylase solution at a concentration of 75 U/mL in ultrapure water (UPW), 0.010 mL of CaCl_2_ at a concentration of 0.3 mol/L, and 0.190 mL of UPW. After undergoing salivary digestion for a period of 2 min, the mixture was subjected to the addition of 3.2 mL of simulated gastric fluid (SGF), along with 0.2 mL of pepsin solution at a concentration of 2000 U/mL in UPW, 0.002 mL of CaCl_2_ at a concentration of 0.3 mol/L, 0.2 mL of HCl at a concentration of 1N (to achieve pH 3), and 0.348 mL of ultrapure water. This gastric mixture was then incubated at a temperature of 37 °C for a duration of 2 h within an incubator equipped with a horizontal shaker operating at 120 revolutions per minute.

Following the gastric digestion phase, intestinal digestion was carried out by adding 3.4 mL of simulated intestinal fluid (SIF) to the mixture, along with 2 mL of pancreatin solution at a concentration of 100 U/mL in SIF, 1 mL of bile solution at a concentration of 10 mM in SIF, 0.016 mL of CaCl_2_ at a concentration of 0.3 mol/L, 0.05 mL of NaOH at a concentration of 1N (to achieve pH 7), and 1.534 mL of UPW. The gastrointestinal mixture was then incubated at a temperature of 37 °C for 2 h within an incubator equipped with a horizontal shaker operating at 120 revolutions per minute.

After the completion of digestion, the samples were cooled using an ice bath for a period of 10 min and subsequently subjected to centrifugation at a force of 7500 times the acceleration due to gravity (g) for 10 min at a temperature of 0 °C. This centrifugation process allowed for the separation of the bioaccessible fraction (found in the supernatant) from the non-bioaccessible fraction (present in the solid portion). The bioaccessible fractions were then extracted using the procedure outlined in Section 2.3.1, employing 4 g of the bioaccessible fraction instead of the 2 g of biscuit used previously.

### 2.4. Liquid Chromatography-Mass Spectrometry Parameters

Mycotoxins analysis was performed using a high-performance liquid chromatography (HPLC) system Agilent series 1290 RRLC instrument (Agilent, Santa Clara, CA, USA) equipped with a binary pump (G4220A), an autosampler thermostat (G1330B), and a column compartment thermostat (G1316C). The HPLC was coupled to an Agilent triple quadrupole mass spectrometer (6460 A) with a Jet Stream electronic spray ionization (ESI) source (G1958-65138). A Zorbax Eclipse plus C18 column (100 × 2.1 mm, 1.8 μm particle size) from Agilent (San Jose, CA, USA) was used as the analytical column. Gradient elution was performed using a mobile phase composed of an aqueous phase (water/methanol/acetic acid, 94:5:1 (*v*/*v*/*v*), and 5 mM ammonium acetate) and an organic phase (methanol). The gradient elution started at 95% of A and linearly decreased to 65% in 3.5 min. From 3.5 to 4.5 min, A was decreased to 35%, from 4.5 to 8.5 min to 25%, and from 8.5 to 9.5 min to 0% (hold 0.5 min). Finally, it was returned to the initial condition (95% A) in 1 min and kept constant for 2 min. The flow rate was 0.3 mL/min, the column temperature was set at 25 °C, and the total running time was 13 min.

The MassHunter software (Agilent) was used during the optimization and quantification stages. In Table 2, the precursor ion, the product ions, and the optimized energies for mycotoxins are shown.

### 2.5. Analytical Method Validation

The analytical method was validated according to the parameters determined by the SANTE guidelines [31]. Linearity, matrix effect, accuracy, precision (intra-day and inter-day precision), and limits of detection (LODs) and quantification (LOQs) were estimated in biscuits and bioaccessible fractions (Table 3).

The linearity and matrix effects were determined using solvent calibration curves (ACN) and blank extraction matrix calibration curves. Linearity, defined as R^2^, was between 0.9899 and 0.9994. Despite the addition of the cleaning step, the matrix effect was significant, and for DON (−610 to −520), strong signal suppression was observed.

Recoveries (expressed as %) and intra- and inter-day precision (expressed as relative standard deviation (%RSD)) values were established at concentration levels of LOQs and 500 µg/kg. Recoveries ranged from 68 to 120%, and precision values were lower than 20% (Table 3).

LOQs were determined as the lowest concentration that provided acceptable values of recoveries and precision (between 70 and 120%), and LODs were determined as the lowest concentration of the signal-to-noise ratio was lower than 3.

### 2.6. Color Measurements

The color of the biscuits was evaluated in terms of fiber enrichment using the L*a*b system, where L* represents the luminosity or lightness component, a* represents the intensity of red (+) and green (−), and b* represents the intensity of yellow (+) and blue (−). A colorimeter (Chroma Meter, CR-400, Konica Minolta, Tokyo, Japan) that had been calibrated with a white standard tile was used to analyze the biscuits based on the aforementioned parameters. Color distance (ΔE), a dimensionless parameter that reflects the combination of L*a*b values when sample pairs are compared, was determined using Equation (1). The difference in biscuit color between pairs of samples was quantified using the ΔE value, which determined whether a perceptible difference in color was present, considering specific thresholds. Specifically, when ΔE is less than 1, it signifies a difference that is typically imperceptible to the human eye. When ΔE falls between 1 and 2, a very slight difference can be discerned, primarily by a trained observer. For ΔE values between 2 and 3.5, a moderate difference is apparent, even to an untrained eye. In cases where ΔE ranges from 3.5 to 6, the difference becomes noticeable, and when ΔE exceeds 6, the distinction becomes significantly pronounced [32]. The color data provided in this study are an average derived from 16 measurements taken from 4 individual biscuits within each formulation.
(1)$$∆E=\sqrt{{\left(L_{2}-L_{1}\right)}^{2}+{\left(a_{2}-a_{1}\right)}^{2}+{\left(b_{2}-b_{1}\right)}^{2}}$$

### 2.7. Statistical Analysis

The differences between samples were analyzed by statistical analysis (*t*-test) using the Microsoft Office Excel 365 statistical software package and the T Score to *p* Value Calculator (https://www.statology.org/t-score-p-value-calculator/ (accessed on 15 May 2023)).

### 2.8. UHPLC-Q-Orbitrap-MS for Non-Targeted Analysis

The analysis was conducted using Thermo Fisher Scientific Vanquish Flex Quaternary LC (Thermo Fisher Scientific TranscendTM, Thermo Fisher Scientific, San Jose, CA, USA) combined with a Q-Exactive Orbitrap hybrid mass spectrometer, both from Thermo Fisher Scientific (Exactive^TM^, Thermo Fisher Scientific, Bremen, Germany). It also uses a heating electrospray interface (HESI-II, Thermo Fisher Scientific, San Jose, CA, USA) in positive and negative modes. Separation was carried out under the same chromatographic conditions as explained in Section 2.4.

The ESI parameters used were spray voltage at 4 kV; sheath gas (N2, 95%), 35 (arbitrary units); auxiliary gas (N2, 95%), 10 (arbitrary units); S-lens RF level, 50 (arbitrary units); capillary temperature, 300 °C; and heater temperature, 305 °C. The mass spectra were acquired using four different acquisition functions: (1) full MS, ESI+, without fragmentation (the higher collisional dissociation (HCD) collision cell was switched off), mass resolving power = 70,000 Full Width at Half Maximum (FWHM); AGC target = 1 × 10^6^; (2) full MS, ESI-, without fragmentation (the higher collisional dissociation (HCD) collision cell was switched off), mass resolving power = 70,000 FWHM; AGC target = 1 × 10^6^; (3) DIA-MS/MS, ESI+ (HCD on, collision energy = 30 eV), mass resolving power = 35,000 FWHM; AGC target = 2 × 10^5^; and (4) DIA-MS/MS, ESI- (HCD on, collision energy = 30 eV), mass resolving power = 35,000 FWHM; AGC target = 2 × 10^5^. The full scan-MS mode ranged between m/z 60 and 900.

The chromatograms were obtained using external calibration mode and subsequently processed with Xcalibur™ version 4.3.73, with Quan and Qual Browser, and a homemade database created with data reported about metabolites of mycotoxins.

## 3. Results and Discussion

### 3.1. LC-QqQ-MS Optimization

First, MS characterization was carried out by direct infusion of standard solutions of the analytes at 10 mg/L in acetonitrile (ACN) at a flow rate of 0.15 mL/min. Compounds were analyzed using ESI+ and ESI-. Full scan and MS/MS spectra were acquired to obtain the most sensitive transitions. Further optimization was performed to evaluate the intensity of precursor ions obtained from different fragmentor voltages (from 75 to 200 V) and collision energies (CE, from 10 to 50 eV) for each product ion. The characteristic precursor and product ions are described in Section 2.4.

Chromatographic characterization was performed using conditions previously developed in the research group for mycotoxins [33]. The analytical column used was Zorbax Eclipse plus C18, and the mobile phase was composed of an aqueous phase (water/methanol/acetic acid, 94:5:1 (*v*/*v*/*v*) and 5 mM ammonium acetate) and an organic phase (methanol). Gradient elution was optimized to reduce the analysis time and improve the elution of the analytes, beginning with a gradient of 25 min [33] to obtain an optimized gradient of 13 min, modifying the changes in the percentages of water at the beginning from 95-35 (in 7 min) to 95-65-35 (in 4.5 min) and the step of 100% of organic solvent (hold) from 3 min to 0.5 min. The flow rate was also optimized at 0.3 mL/min, obtaining a good peak shape for all compounds. Figure 1 shows the extracted ion chromatograms for the analytes at the optimized retention time.

### 3.2. Evaluation of Dietary Fibers as Blinders to Mycotoxins

To test the binding of mycotoxins to dietary fibers, five types of sources were employed: arabinogalactan (A), k-carrageenan (K), commercial pectin (PC), apple pomace flour with sugars (FS), and powered apple pomace flour without sugar (FP). PC and K are negatively charged, while A is neutrally charged. PC and A are plant-based polysaccharides. Pectin is present in a large diversity of fruits but is normally obtained from apples and consists of linear units of (α1→4)-D-GalpA residues [34]. It has attracted much attention as a widely used natural food additive for the preparation of jams, jellies, and marmalades [35]. FS and FP are a type of apple flour rich in pectin obtained as a by-product during apple juice production, and their preparation is indicated in Section 2.1. A is derived from larch wood and made up of a branched biopolymer composed mainly of a 1,4 linked β-D-Galp backbone with α-L-Araf residues. K is a linear sulfated mucopolysaccharide found in the cell walls of red algae.

To evaluate the impact of these five fibers (A, K, PC, FS, FP) on mycotoxins released during gastrointestinal digestion, biscuit recipes were prepared following the procedure described in Section 2.2; the final dough was spiked at a known concentration (500 µg/kg of each mycotoxin). The content of soluble fiber (pectin) in the flours of apple pomace was determined using a Megazyme fiber test to determine the total content of flour that must be added to obtain a pectin content in the biscuits compared with PC (5% of pectin) (Appendix A). In the pectin case of FS, the content in the flour was 7.2%, whereas for FP it was 15.9%.

Once the samples were prepared, they were treated as indicated in Section 2.3.1 and the content of mycotoxins was assessed as described in Section 2.4. First, concentrations without the digestion process were determined (Table 4). According to previous studies, all studied mycotoxins are stable during baking processes [36,37], which is confirmed in this study. For example, DON is slightly degraded after 40 min at 210 °C, and ZEA was reduced only after 60 min at 220 °C. In the case of HT-2 and T-2 toxins, their stability is lower at high temperatures, and according to the study of Sören Kuchenbuch et al., 45% of T-2 and 20% of HT-2 were thermally degraded, depending on the water content, baking time, and temperature [38]. However, the highest percentages of degradation were obtained at a tested temperature of 200 °C for 30 min, while in the present article, biscuits were made for 12 min at 190 °C.

No significant effects of the dietary fiber occurred for DON, ZEA, HT-2, and T-2 during the preparation process, except in the case of ZEA for K biscuits, where a reduction of 40% of the content of ZEA compared to control samples was observed (Table 4). The mitigation of ZEA is due to K entrapped ZEA in the fiber matrix because according to literature k-carrageenan form gels at pH 8 (the pH of the biscuits was 8.2) and in the presence of K^+^ ions [39]. The other fibers, such as pectin, form gels at pH 2–3.5, and the protonation of carboxylic groups is important for strong hydrogen bonds. At the pH of the biscuits (8.2), pectin does not form gels, and in addition, Ca^2+^ ions are required for gelification. Arabinogalactans form stable emulsions at pH 2–8, increasing the pH decreases the stability of the emulsions, so it was probable that they are not stable at pH 8.2.

In addition, to determine the impact of fiber enrichment on the doneness and appearance of the cooking of the enriched biscuits compared to the conventional ones (control), the color measure of the biscuits was performed by L∗, a∗, and b∗ as indicated in Section 2.6. The measurement showed that the addition of A and K to the biscuits promoted medium differences compared to the control biscuits (ΔE = 2.86 and 3.41, respectively), while for the biscuits containing PC or FS, the difference in color was obvious (ΔE = 4.30 and 5.25, respectively). For FP, the difference was very obvious (ΔE = 11.93) (Appendix A). Figure 2 shows the appearance of the baked biscuits with the different fibers used in the study. As can be observed, some color differences were observed in the biscuits enriched with fiber when compared with the control.

### 3.3. Bioaccessibility of Mycotoxins after In Vitro Digestion of Biscuits

In vitro gastric and gastrointestinal digestion was performed following a standardized method (INFOGEST) described in Section 2.3.2. Bioaccessibility was calculated in each phase using Equation (2).
(2)$$Bioacessibility\left(\%\right)=\left(\frac{Concentration\;in\;digested\;extract}{Concentration\;in\;biscuit}\right)\times 100$$

The bioaccessibility values represent the percentage of each compound that is available to be absorbed in each phase and assume the free fraction of the matrix (concentration of mycotoxins in biscuits (Table 4)). Normally, the most relevant bioaccessibility results are those from the complete stage (gastrointestinal phase) because of its complete information on absorption. However, the studied mycotoxins have fast absorption times (e.g., DON is absorbed between 1 and 4 h, and the maximum plasma concentration is reached 0.33 h after ingestion), so it is expected that a percentage of them is absorbed by gastric cells. Indeed, some studies determined through in vitro absorption experiments that some of these compounds can be highly transported across the gastric epithelia, such as, for example, DON, which is absorbed similarly by gastric and by intestinal cell monolayers [40]. Therefore, it was decided to evaluate both phases: gastric and gastrointestinal phases.

Regarding the influence of dietary fibers on gastric bioaccessibility of mycotoxins, the most important reduction occurred in the case of ZEA, where bioaccessibility percentages were lower than 7% in all cases, compared to control samples, where it was 25%. The lowest value was obtained using K-carrageenan (Figure 3). For DON, HT-2, and T-2, even though a reduction was observed, the bioaccessibility results were higher than 60% (Figure 3). The fiber most suitable for these three mycotoxins was FP, with bioaccessibility values of 72%, 88%, and 67% for DON, HT-2, and T-2, respectively.

For the gastrointestinal phase, the bioaccessibility of ZEA increased with respect to the gastric phase, obtaining values around 20–40%, while the control value was 60%. In this case, as occurred in the gastric phase, the fiber that provided the lowest results of bioaccessibility was K-carrageenan (20%). For DON, bioaccessibility also increased with respect to the gastric phase, obtaining the lowest bioaccessibility results employing FP (78%), as in the gastric tract. FP was composed primarily of pectin, and, as Tamura et al. indicated in their study, pectin trapped the mycotoxin DON in its “egg box” and significantly reduced its bioaccessibility [41]. Finally, for HT-2 and T-2, a low reduction in bioaccessibility was observed with respect to the gastric phase, providing similar results to the dietary fibers of apple pomace (FP and FS). Bioaccessibility values were lower in the case of T-2, around 50%, than in the case of HT-2 (80%). These results agree with De Angelis et al. who evaluated the bioaccessibility of HT-2 and T-2 in fortified and naturally contaminated bread, obtaining a bioaccessibility for HT-2 at around 95%, while for T-2 it was around 45% [42].

In addition to that, a *t*-test statistical analysis was carried out to observe differences between biscuits enriched with dietary fibers and control biscuits, observing in all cases *p*-values lower than 0.05, indicating significant differences (Table 4).

In conclusion, it should be noted that ZEA is the mycotoxin most affected by a decrease in bioaccessibility after in vitro digestion and with the use of dietary fibers. Differences are observed for other mycotoxins, but they are not as pronounced as they are for ZEA. The mitigation of ZEA could be similar to what occurred with other mycotoxins in previous works, where the use of fibers could be responsible for naturally absorbing materials. Dietary fibers, which could combine some bioactive compounds such as mycotoxins or polyphenols, can reduce their percent of bioaccessibility [39]. This could result from linear fibers that trap the ZEA molecule in an egg-box structure and avoid its availability.

Finally, the inclusion of dietary fibers has been shown to protect against toxicoses resulting from numerous xenobiotic compounds [43,44] and can be applied in food and feed as a cost-effective method of detoxifying them from mycotoxins. Among the promising dietary fibers for reducing mycotoxin content, we can highlight that K-carrageenan for ZEA and FP for the rest of the compounds are the best fibers.

### 3.4. Non-Targeted Analysis for Metabolite Search

In addition to the mycotoxins monitored during in vitro digestion, and bearing in mind that some of their metabolites could be present in extracts as conjugates, such as glucoside-type, a list of 30 potential conjugates collected from previous bibliographic data [45,46,47,48,49] were used to create a home database. Using a non-targeted strategy, the samples were acquired in full scan mode, with data independent mass spectrometry fragmentation (DIA-MS/MS), and using the power of the exact mass of high-resolution mass spectrometry (HRMS) to detect the largest number of compounds present in the samples. The identification criteria for positive potential compounds were mass error lower than 5 ppm and detection of at least two ions (precursor + fragment ion).

Tentative results indicate that three metabolites of the studied mycotoxins, zearalenone-14-glucoside, HT-2-3-glucuronide, and T-2-triol, were present in the in vitro phases. Due to the absence of analytical standards, a semi-quantification was carried out, and their concentrations were calculated with matrix match calibration curves of the parent compounds (Figure 4). For all of them, the same differences were observed in their appearance when different dietary fibers were employed, as occurred with the parent compound, observing the presence of glucose conjugates in major proportions when apple flour was used. In addition, differences can be observed in the gastric and the gastrointestinal phases, with higher concentrations in the gastrointestinal tract for the three metabolites.

Concentrations of HT-2-3-glucuronide and Zearalenone-14-glucoside represent 30% and 10% of the total content of the respective parent compounds (HT-2 and ZEA); however, the concentration of T-2-triol was really higher with respect to the parent compound concentration and represented around 500% of the total content of T-2 in the gastric phase and 2000% in the gastrointestinal phase. Therefore, and taking into account that the amount of metabolite generated from the parent cannot be higher than the amount of parent, it is expected that T-2-triol could previously be in the samples or its quantification was not precise due to it being quantified with the standard of T-2. Despite that, this metabolite is important because T-2 metabolizes to it during in vitro digestion and must be taken into consideration for its toxicity. This was also previously reported for the hydroxylated metabolite detected at a higher rate in urine samples [46].

## 4. Future Perspectives and Conclusions

This study offers new insights into the reduction and mitigation of mycotoxin levels in biscuits using dietary fibers. The color test demonstrated that significant differences are presented when dietary fibers are added to the biscuits. The results obtained after the in vitro digestion test indicated that the addition of dietary fibers significantly decreased mycotoxin bioaccessibilities compared to the control samples. K-carrageenan seems to be the most effective fiber for ZEA bioaccessibility reduction, while FP worked best for DON, HT-2, and T-2. These two fibers adsorb mycotoxins, reducing their bioavailability, and could lead to a decrease in the toxic effects of mycotoxins and potentially reduce the risk of adverse health effects. These findings are valuable for the food industry as they provide new ways to reduce mycotoxin levels in food products. Additionally, the use of apple flour was derived from apple pomace, which is considered a bio-circular economy product and a cost-effective solution obtained during apple juice production. Additionally, the addition of this dietary fiber to the biscuits could increase their fiber content, making them nutritionally better than the standard recipe.

Non-targeted analysis reveals the appearance of three metabolites of mycotoxins in in vitro digestion phases, showing that in gastrointestinal phases their presence is higher. Special attention is necessary for the metabolite T-2 triol due to the high rate of concentration detected.

The next steps related to this work can be focused on the identification of additional dietary fibers with higher reduction effectiveness than the studied ones, exploring their mechanisms of action. As it is obtained here, the addition of certain dietary fibers can reduce mycotoxin levels in biscuits. Therefore, another important point to improve in the future is the development of food products with reduced mycotoxin levels through the addition of certain additives such as fibers, including frequently contaminated food products, such as bread or pasta. This may lead to a reduction of the risk of adverse health effects associated with mycotoxins. A final point to take into consideration is the development of healthier food options through, for example, the addition of dietary fibers. In addition, the use of fibers coming from by-products such as those used in this study could improve resource uses and achieve a greener food industry. Therefore, the source of new fiber alternatives should be investigated.

## Figures and Tables

**Figure 1 foods-12-03175-f001:**
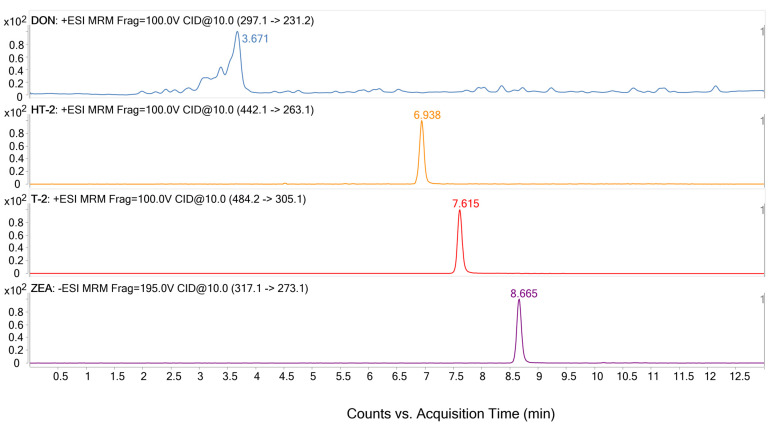
Extracted ion chromatograms of the analytes studied at 0.2 mg/L in ACN.

**Figure 2 foods-12-03175-f002:**
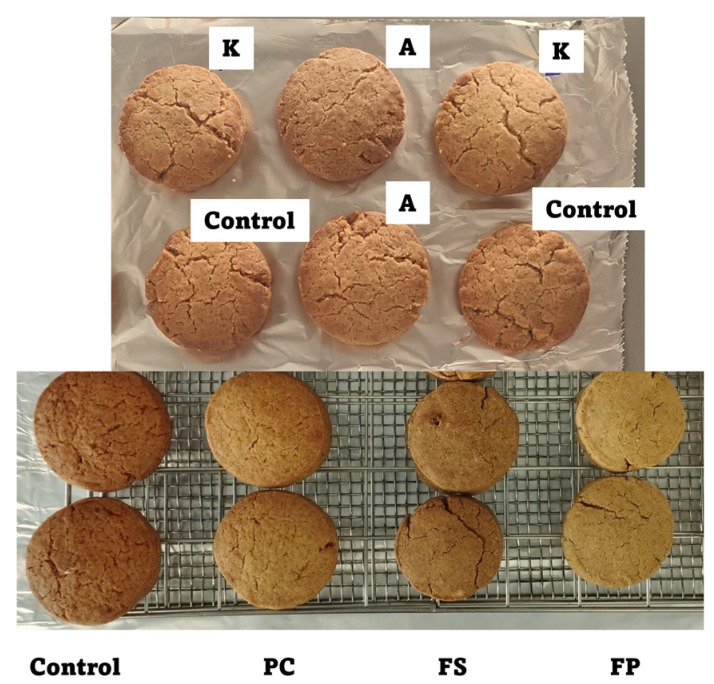
Appearance of control biscuits and biscuits enriched with dietary fibers (A: arabinogalactan; K: k-carrageenan; PC: commercial pectin; FS: apple pomace flour with sugars; and FP: powered apple pomace flour without sugar).

**Figure 3 foods-12-03175-f003:**
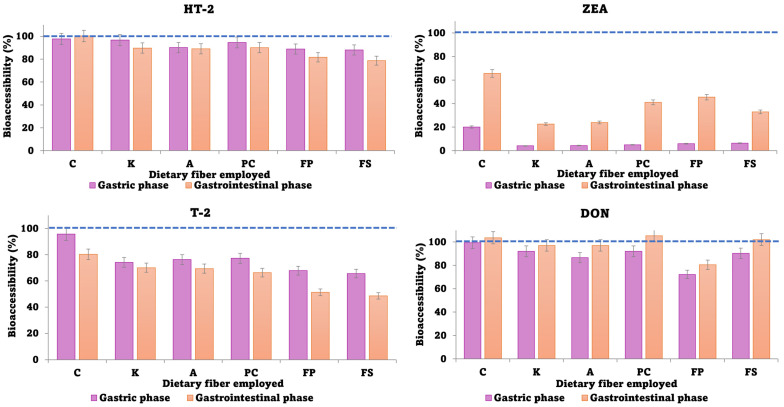
Bioaccessibility after the in vitro digestion in the gastric and gastrointestinal phase. (Error bars *n* = 3; A: arabinogalactan; C: control; K: k-carrageenan; PC: commercial pectin; FS: apple flour with sugars; and FP: powered apple flour without sugar). Note: the line represents the initial content of mycotoxins in the cooked biscuits (100%), except for ZEA using K (60%).

**Figure 4 foods-12-03175-f004:**
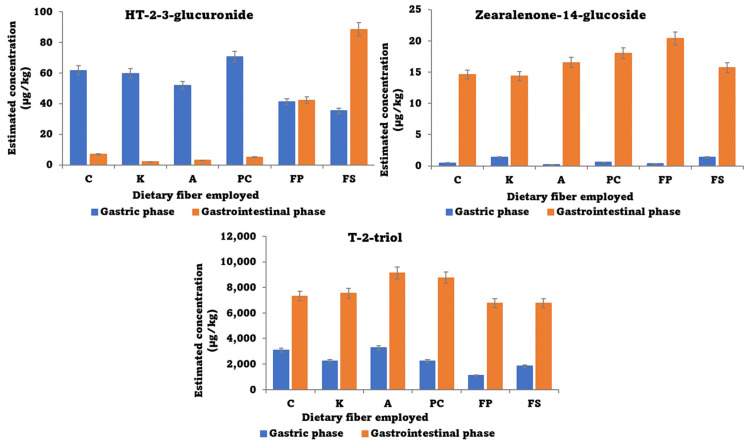
Metabolites estimated concentration after the in vitro digestion in the gastric and gastrointestinal phase (µg/kg) (Error bars *n* = 3; A: arabinogalactan; C: control; K: k-carrageenan; PC: commercial pectin; FS: apple flour with sugars; and FP: powered apple flour without sugar).

**Table 1 foods-12-03175-t001:** Total amount of ingredients used in the formulation of biscuits using 5% of dietary fibers (related to the amount of flour).

Ingredients	Recipe with A, K, PC	Recipe with FS	Recipe with FP
**Flour**	475 g	150 g	345 g
**NH_4_HCO_3_**	2 g	2 g	2 g
**NaHCO_3_**	9 g	9 g	9 g
**Sugar**	136 g	-	136 g
**Salt**	4 g	4 g	4 g
**Malt extract**	20 g	20 g	20 g
**Glucose-fructose syrup**	100 g	-	100 g
**Sunflower oil**	117 g	117 g	117 g
**Water**	75 mL	185 mL	175 mL
**Dietary fiber**	25 g	350 g	155 g

Abbreviations: A: arabinogalactan; K: k-carrageenan; PC: pectin commercial; FS: apple flour with sugars; and FP: apple flour without sugar powered.

**Table 2 foods-12-03175-t002:** LC-MS/MS parameters for mycotoxins.

Analyte	Precursor Ion (m/z)	Product Ion (m/z)	Cone Voltage (V)	Collision Energy (eV)	Ionization Mode	Retention Time (min)
DON	297.1	249.1	100	10	Positive	3.67
		231.2	100	10		
HT-2	442.1	263.1	100	10	Positive	6.93
		214.9	100	10		
T-2	484.2	305.1	100	10	Positive	7.62
		215.1	100	10		
ZEA	317.1	273.1	195	10	Negative	8.66
		175.0	195	20		
^13^C_15_DON	310.1	278.8	150	10	Negative	3.11
		250.3	150	10		

**Table 3 foods-12-03175-t003:** Validation results for the proposed method in biscuits and bioaccessible fractions.

Compounds		Linearity	Matrix Effect ^a^	Recoveries (%)		Intraday Precision ^c^ (% RSD)		Interday Precision ^c^ (% RSD)		LOQ (µg/kg)	LOD (µg/kg)
				LOQ ^b^	500 µg/kg	LOQ ^b^	500 µg/kg	LOQ ^b^	500 µg/kg		
DON	B	0.9950	−610	79	85	12	8	16	10	50	25
	GP	0.9918	−520	82	89	10	9	20	15	100	50
	GIP	0.9955	−555	65	69	18	15	17	12	100	50
ZEA	B	0.9994	31	118	120	4	2	3	2	2.5	0.5
	GP	0.9965	40	102	102	5	4	4	3	10	2
	GIP	0.9945	65	91	96	8	5	8	4	10	2
HT-2	B	0.9951	−6	72	95	6	8	8	5	7.5	2.5
	GP	0.9900	−15	78	78	12	9	9	6	10	5
	GIP	0.9917	−26	82	85	10	11	11	7	10	5
T-2	B	0.9980	−14	71	85	5	3	3	2	2.5	0.5
	GP	0.9899	−5	68	68	10	5	5	3	10	2
	GIP	0.9902	−18	85	96	9	6	6	1	10	2

^a^ Equation used: $Matrix\;effect=\left(\frac{slope\;in\;matrix}{slope\;in\;solvent}-1\right)\times 100$; ^b^ parameter estimated at the LOQ of the analyte; ^c^
*n* = 5; abbreviations: B: biscuits; GP: gastric phase; and GIP: gastrointestinal phase.

**Table 4 foods-12-03175-t004:** Concentration of mycotoxins in biscuits spiked at 500.0 µg/kg and after an in vitro digestion procedure (*n* = 3, relative standard deviation (RSD) in % between parenthesis).

	Analytes			
Dietary Fibers	DON	HT-2	T-2	ZEA
**Biscuits Concentration µg/kg**				
**C**	497.0 (5)	507.0 (5)	504.0 (3)	502.3 (2)
**K**	489.3 (6)	490.3 (9)	495.0 (2)	295.0 (3)
**A**	502.1 (5)	488.0 (12)	490.2 (13)	475.0 (15)
**PC**	501.3 (2)	514.2 (14)	511.3 (10)	499.3 (1)
**FP**	499.2 (1)	495.3 (4)	505.2 (7)	498.0 (5)
**FS**	500.9 (3)	508.4 (10)	501.4 (1)	490.0 (4)
**Gastric Phase, Concentration µg/kg**				
**C**	496.7 (2)	488.2 (18)	478.4 (9)	100.7 (1)
**K**	460.2 (5)	483.2 (6)	370.9 (10)	20.4 (3)
**A**	433.0 (8)	450.6 (8)	381.0 (10)	30.9 (5)
**PC**	460.2 (12)	472.9 (12)	385.9 (13)	25.1 (5)
**FP**	360.9 (10)	444.3 (13)	338.8 (4)	29.4 (4)
**FS**	451.1 (8)	439.8 (5)	328.0 (8)	35.6 (3)
** *p* ** **-value ^1^ **	0.00542	0.00468	0.00009	0.00000
**Gastrointestinal Phase, Concentration µg/kg**				
**C**	518.3 (13)	501.1 (5)	401.3 (7)	329.9 (12)
**K**	485.4 (5)	448.5 (6)	311.2 (6)	113.0 (5)
**A**	485.2 (7)	465.1 (8)	346.6 (9)	169.4 (6)
**PC**	527.1 (8)	503.4 (10)	331.5 (5)	205.3 (10)
**FP**	402.2 (11)	408.3 (4)	256.3 (10)	226.4 (8)
**FS**	510.3 (4)	393.2 (3)	242.6 (5)	185.4 (7)
** *p* ** **-value ^1^**	0.04470	0.00928	0.00131	0.00025

Abbreviations: A: arabinogalactan; C: Control; K: k-carrageenan; PC: commercial pectin; FS: apple pomace flour with sugars; and FP: powered apple pomace flour without sugar. ^1^ *p*-value obtained from a *t*-test statistical analysis (control vs. enriched samples for each mycotoxin).

## Data Availability

The data presented in this study are available in this article and in the Appendix A.

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
