# Peer review of "Investigating the Impact of Dietary Fibers on Mycotoxin Bioaccessibility during In Vitro Biscuit Digestion and Metabolites Identification"

_foods, 2023, doi:10.3390/foods12173175_

Round 1

Reviewer 1 Report

I have reviewed the manuscript entitled: Investigating the impact of dietary fibers on mycotoxin bioaccessibility during in vitro biscuit digestion and metabolites identification

It is a very interesting article that addresses a very important topic: Reducing bioavailability of mycotoxins. Also the reuse of by-products contributes to health through a final product. The information is correct, well discussed, with excellent results and a well-detailed methodology. The information presented in this investigation is relevant considering the problematic and the  searching for alternatives that could reduce the presence of mycotoxins in foods.

Anyway I have some recommendations.

Introduction: ok

Material and Methods

Line 107-126. This part is very messy, lacks order, it is suggested to separate by subtitles, some methodologies mentioned in this section.

Section 2.2: In the methodology section 2.2 is not mentioned that the muffins were inoculated with Mycotoxins, however it is mentioned in the results, please clarify it in the methodology.

Results and Discussion section

In the results section, it is suggested that you keep the sequence mentioned in the methodology, since some results are mentioned in different order and the sequence is not understood.

 Line 425-436.

Conclusions are mentioned in this section, it is suggested to remove this part and add it in the conclusions section.

Author Response

Response to Reviewer 1 Comments

Point 1: Line 107-126. This part is very messy, lacks order, it is suggested to separate by subtitles, some methodologies mentioned in this section.

Response 1: A new section (Section 2.1.1.) was added that explained the dietary fibres used for biscuits preparation

Point 2: Section 2.2: In the methodology section 2.2 is not mentioned that the muffins were inoculated with Mycotoxins, however it is mentioned in the results, please clarify it in the methodology.

Response 2: A clarification was added in the methodology section.

Point 3: In the results section, it is suggested that you keep the sequence mentioned in the methodology, since some results are mentioned in different order and the sequence is not understood.

 Response 3: Some parts of the results sections were reorganised according to reviewer suggestion

Point 4: Line 425-436. Conclusions are mentioned in this section, it is suggested to remove this part and add it in the conclusions section.

 Response 4: The authors decided to keep these paragraphs in this section because they compared the results with other studies from bibliography, and in the conclusions section only conclusions to the results obtained in this work are included.

Reviewer 2 Report

Authors should include and explain the following comments in the revised manuscript

1.     Mycotoxins’ Analytical Method Performance

2.     Fortification Levels of Zearalenone Analysis

3.     Bioaccessibility of Zearalenone in Studied Biscuits

4.     Availability of Zearalenone in Salival, Gastric, and Intestinal Phase in Biscuits

5.     Impact of Bioaccessible Mycotoxins on Intestinal Cells’ Viability and Inflammation

6.     In vitro Digestion and Bioaccessibility should include in a separate column

7.     Intestinal Absorption Assay

 Reconsider after major revision

Author Response

Response to Reviewer 2 Comments

Point 1: Mycotoxins’ Analytical Method Performance

 Response 1: The validation results for the analytical method of mycotoxins are described in Table 3, where recoveries (performance) were indicated in the table.

Point 2: Fortification Levels of Zearalenone Analysis

Response 2: The Fortification levels of zearalenone were the same as those of the other mycotoxins (DON, HT-2 and T-2) and the explanation was to try to have a concentration of enoght that could be dectectable after in-vitro digestion process and to have the capacity to detect metabolites by non-targeted approaches.

Point 3: Bioaccessibility of Zearalenone in Studied Biscuits

Response 3: The bioaccesibility of all mycotoxins was discussed in Section 3.3

Point 4: Availability of Zearalenone in Salival, Gastric, and Intestinal Phase in Biscuits

Response 4: If the reviewer refers to bioavailability, these parameters are only possible to measure by in vivo studies or cell assays, and it is not the goal of this work, that it is the use of an in vitro digestion protocol followed by a standardised method previously developed as indicated in the manuscript. In this standarized method, bioaccesibility in gactric and gastrointestinal phase was evaluated as indicated in the manuscript.  

Point 5: Impact of Bioaccessible Mycotoxins on Intestinal Cells’ Viability and Inflammation

 Response 5: How I said before, the goal of the work is the evaluation of the bioaccesibility, not the evaluation of Cells’ Viability or citotoxicity assays; for this reason the mentioned impact was not evaluated; in addition as previous literature reported, mycotoxins, as expected, are toxic for the human body. In some parts of the manuscript this information was indicated.  

Point 6: In vitro Digestion and Bioaccessibility should include in a separate column

Response 6: The authors do not understand the question of the reviewer. Bioaccessibility is the parameter obtained which allows us to understand how in vitro digestion has occurred, so they are the same thing

Point 7: Intestinal Absorption Assay

Response 7: According to response 5, this is not the goal of the study.